# A new QRS detector stress test combining temporal jitter and F-score (JF) reveals significant performance differences amongst popular detectors

**Bernd Porr** [1]*, **Peter W. Macfarlane** [2]

**1** School of Engineering, Biomedical Engineering, University of Glasgow, Glasgow, United Kingdom,
**2** Electrocardiology Group, Institute of Health and Wellbeing, University of Glasgow, Glasgow, United Kingdom

\* bernd.porr@glasgow.ac.uk

## Abstract

QRS detection within an electrocardiogram (ECG) is the basis of virtually any further processing and any error caused by this detection will propagate to further processing stages. However, standard benchmarking procedures of QRS detectors are seriously flawed because they report almost always close to 100% accuracy for any QRS detection algorithm. This is due to the use of large temporal error margins and noise-free ECG databases which grossly overestimate their performance. The use of a large fixed error margin masks temporal jitter between detection and ground truth measurements. Here, we present a new performance measure (JF) which combines temporal jitter with the F-score, and also an ECG database with decreasing levels of signal to noise ratios based on noise generated from different tasks. Our new performance measure JF fully encompasses all the types of errors that can occur, equally weights them and provides a percentage value which allows direct comparison between QRS detection algorithms. In combination with the new noisy ECG database, the JF performance measure now varies between 50% and 100% for different detectors and signal to noise conditions thereby making it possible to find the best detector for an application.

**Data Availability Statement:** All data files are available from: Howell, L., & Bernd Porr. (2024). High precision ECG Database with annotated R peaks (1.1.0). Zenodo. https://doi.org/10.5281/zenodo.10925903.

## Introduction

An electrocardiogram (ECG) records the electrical activity of the heart [1]. In the normal heart, each cardiac cycle (heartbeat) consists of atrial activation which produces a P wave. This is followed by atrial repolarisation, and ventricular activation which gives rise to the QRS complex. Atrial repolarisation is generally hidden by the QRS complex. Finally, ventricular repolarisation follows ventricular activation giving rise to a T wave. In an abnormal recording, the P wave may not be present though other signs of atrial activity may be seen. In some abnormal cardiac cycles, the source of ventricular activation may be in the ventricles giving one or more QRS complexes of a different morphology compared to the normally conducted beat for that

**Funding:** The authors received no specific funding for this work.

**Competing interests:** B.P. is CEO of Glasgow Neuro LTD which manufactures the Attys DAQ board. He has received a salary from Glasgow Neuro LTD which, however, did not provide any funding for this study. B.P. was involved in the design, data analysis, decision to publish, and preparation of the manuscript. However, this study is not about comparing different ECG measurement devices. The outcome of this study will be identical with other commercial data acquisition devices with identical specifications and the Attys does not alter the originally measured data in any way and thus does not introduce any bias. This does not alter our adherence to PLOS ONE policies on sharing data and materials. All data is available publicly and in its unprocessed format as measured by the Attys. There are no patents, products in development, or marketed products associated with this research to declare.

heart. In all cases, the most identifiable wave with the fastest rate of inscription is the QRS complex [2]. Detecting the QRS complex is the first step in calculating any further measures. Most QRS detection algorithms will use a distinctive R peak as may be found in one or two lead recording with carefully placed electrodes but if more leads are available, a function which combines leads and automatically produces a prominent R wave can be used [3].

When the presence of a QRS complex has been detected, further measures such as heart rate and heart rate variability (HRV) can be calculated. Heart rate can simply be calculated by taking the difference between two QRS detection time stamps, which is mathematically the same as finding the first derivative. However, derivatives amplify noise and thus any errors in the QRS detection will result in even larger errors as further derivatives are taken. When calculating HRV, the 2nd derivative between successive QRS complexes is used, and thus any small errors in the QRS detection will compound into much larger errors in any subsequent HRV result [4]. As such, a heart beat detector algorithm that can accurately and precisely identify the QRS complexes in an ECG, is vital in obtaining a useful and representative HRV. Errors that can occur from poor QRS detector algorithms are: missed QRS complexes (false negatives), extra QRS complexes (false positives), and temporal inaccuracies (temporal jitter). For that reason it is imperative to be able to evaluate detectors and select one which has the lowest occurrence of the above errors. In order to benchmark detectors, the principal requirements are:

1. i) a precisely annotated ECG database with various noise levels and QRS morphologies, and

2. ii) a performance measure taking into account all relevant errors reporting meaningful values to be able to compare detectors.

The current state of the art neither provides the required database nor the performance measure. This paper outlines the current state of the art regarding these two points.

In order to meet the first requirement, an ECG database with precisely annotated QRS complexes is essential. Almost all published research on heartbeat detection algorithms uses the MIT-BIH Arrhythmia Database (MITDB) [5, 6] for testing. This database contains 48 ambulatory, 30-minute-long, annotated ECG recordings, 25 of which contain less common arrhythmias. The recordings have a sampling rate of 360 samples/sec with an 11 bit resolution over a 10 mV range. Although it has become the standard for detector evaluation, the almost exclusive use of this database poses an issue due to its two main shortcomings:

1. **Very few examples of motion artefacts:** This leads to attempts looking only at sections of the ECG recordings which contain a fair amount of noise. The work by [7–9] tried to highlight noise resilience using only small sections (3...10s) of a few select records. As these noisy sections make up such a small proportion of the database, they have very little impact on the overall results.

2. Inconsistent QRS complex annotations which clearly have a temporal jitter (Fig 1). This poses a serious problem when benchmarking the detectors because it introduces an additional temporal jitter in QRS annotation time-stamps.

Of the current 33 available ECG databases on PhysioNet, only two aim to provide examples of noisy ECG signals. However, both have flaws which limit their usefulness. The MIT-BIH Noise Stress Test Database (NSTDB) [10] features synthesised noisy signals that are not representative of a realistic noisy recording. The other database, Motion Artefact Contaminated ECG Database [11], consists of 27 recordings of just a single subject standing and is not

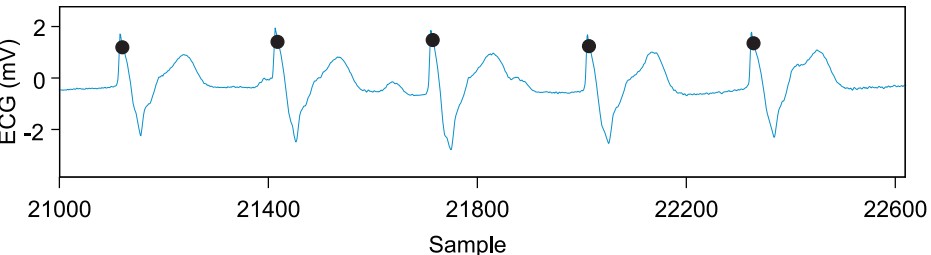

**Fig 1. MITDB heartbeat annotations (dots) not located on R-peaks (record 107) and jitter between annotations.**

annotated, and so is therefore unsuitable for detector evaluation. In conclusion, the current ECG databases are not suitable for QRS detector benchmarking.

After having shown that the current databases are not suitable for detector benchmarking, we now consider the state of the art of performance measures. Currently, the Association for the Advancement of Medical Instrumentation (AAMI) recommends evaluating detector performance on sensitivity (Se), positive predictivity (+P), false positive rate (FPR) and overall accuracy (Acc) [12]. All of these measures are based on the idea that a QRS detection is required to fall in a *fixed* temporal window $w$ which is illustrated in Fig 2. $a_l$ is a set of annotated sample positions where the actual detection has occurred. $d_k$ are the sample positions where the detector has detected the QRS complex. For example, in Fig 2 the first QRS detection has been annotated at $a_0$ and the detection happens at exactly $d_0 = a_0$ the same sample and is classified as a true positive. The 2nd annotation $a_1$ has also a matching QRS complex $d_1$ but this falls outside of the temporal window $|d_1 - a_1| > w$ so that it is a false positive. Of course, there are also cases where there is an annotation as with $a_3$ but no matching QRS detection so that we have a false negative. There will also be detections such as $d_4$ and $d_5$ which are false positives. The standard approach is to count how many detections fall within the temporal window $w$ and count these as "true positives" (TP). With the total number of detections $\text{card}(d_k)$ and annotations $\text{card}(a_l)$ we obtain:

$$TP = \text{card}(|a_l - a_k| < w) \tag{1}$$

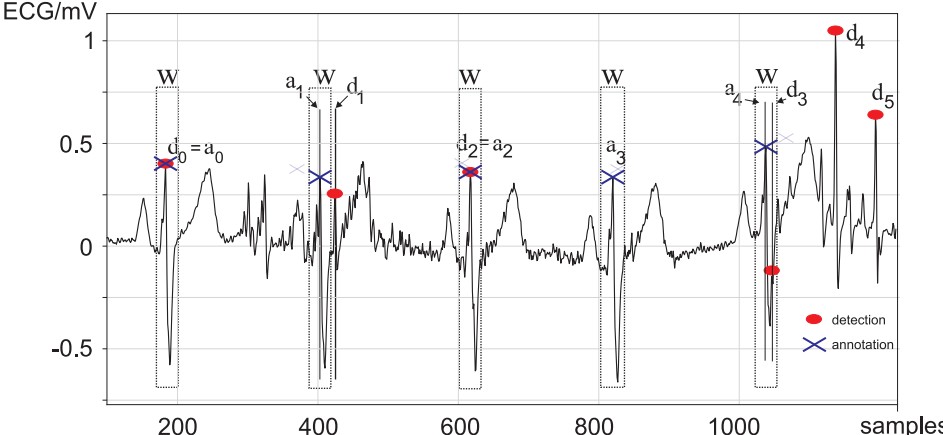

**Fig 2. Examples of ECG detection errors using a fixed temporal window.** $d_{0...4}$ are sample numbers of detected QRS complexes. $a_{0...4}$ are sample numbers of the annotated QRS complex (ground truth). $w$ is the temporal window used for the classical performance measures sensitivity, positive predictivity, false positive rate and overall accuracy.

$$FP \quad = \quad \text{card}(d_k) - TP \tag{2}$$

$$FN \quad = \quad \text{card}(a_l) - TP \tag{3}$$

where card() is the number of elements in a set which is called the cardinal number. For example, if we have three annotations $\{a_0, a_1, a_2\}$ in a set then its cardinal number is: card($\{a_0, a_1, a_2\}$) = 3. $FN$ is the number of false negatives, and $FP$ is the number of false positives where false negatives are the number of missed heartbeats, and false positives are essentially the number of extra detections.

$$SE \quad = \quad \frac{TP}{TP + FN} \tag{4}$$

$$+P \quad = \quad \frac{TP}{TP + FP} \tag{5}$$

$$A \quad = \quad \frac{TP + TN}{TP + TN + FP + FN} \tag{6}$$

where $SE$ is sensitivity, $+P$ is positive predictivity and $A$ the accuracy. Ideally a performance measures should take into account true positives (TP, Eq 1, false positives (FP, Eq 2) and false negatives (FN, Eq 3). However, the sensitivity $SE$ is only taking into account $TP$ and $FN$ while the positive predictivity $+P$ only takes into account $TP$ and $FP$. One might be tempted to use the accuracy but it cannot be calculated because the number of true negatives $TN$ is not known in QRS detection tasks. Thus, all standard measures either miss out on one performance parameter or, in case of the accuracy, cannot be calculated. However, these are not the only problems. The other issue is the fixed temporal window used to calculate $TP$, $FP$ and $FN$ which we discuss now.

The fixed predefined temporal window $w$ of tolerance is widely set as ten times of the sampling interval as advised by Xie and Dubiel [13] in their software library for working with Physionet databases such as the MIT-BIH arrhythmia database [5, 6]:

$$w = \frac{10}{f_s} \tag{7}$$

where $f_s$ is the sampling rate of the ECG recording. For example, at a sampling rate of $f_s$ = 250 samples/sec this gives a temporal window of 40 ms which means that any QRS complex within this window is classified as a true positive and outside of this window as a false negative. Many detector algorithms report high sensitivity scores due to these large windows $w$ [12, 14], yet this is not representative of the actual detector ability across all applications, specifically HRV, as it disregards the temporal jitter. That omitting jitter is problematic can be seen in Fig 2 when comparing the annotations $a_1$ and $a_4$. For $a_1$ the detection $d_1$ lies outside the window $w$ and is classified as a false detection. However, for the annotation $a_4$ the detection $d_3$ lies within the window and is classified as a perfect true positive ignoring the temporal inaccuracy. Thus a resulting high sensitivity might not mean that the detector is perfect if it has a high jitter between detections falling within the window $w$. Gradl et al [15] proposed a possible measure to account for QRS detection inaccuracies using the MITDB. However, their measure uses the absolute deviation from each QRS complex which does not account for detectors that use different detection positions such as the R-S slope [16], and by using the MITDB, errors from

inexact annotations are introduced and are not demonstrated against realistic noisy scenarios. Gradl et al [15] do not attempt to combine their QRS accuracy measure with any of the other possible errors that could occur during detection, and thus there is still no overarching benchmarking parameter proposed.

The combination of current benchmarking parameters not accounting for all error types, large tolerance windows masking temporal inaccuracies, inaccurately annotated ECG recordings, and a lack of realistic noisy recordings, has resulted in all of the most commonly used detector algorithms' performance measures being based on fallacious results under biased conditions. It is clear that the current ECG testing databases and the current recommended performance parameters for benchmarking algorithms are insufficient.

In this paper, we are presenting a new open access ECG database [17] from 25 subjects who performed different tasks such as sitting, performing a maths test, walking on a treadmill, using a hand-bike and jogging: https://doi.org/10.5281/zenodo.10925903. ECGs were simultaneously recorded from both the Einthoven leads and from a chest strap. All ECGs were then annotated at sample precision allowing benchmarking of the different R-peak detectors at the highest possible precision. Having a database with increasing noise levels and strict timing requirements allows us to then determine which detector performs best and highlights the consequences of poor detection. This is particularly relevant for applications such as heart rate variability where 2nd derivative quantities will be most susceptible to R peak jitter. The use of both Einthoven and chest strap tests if detectors are robust against different QRS complex morphologies and give important insights into the applicability of these detectors for chest worn heart rate monitors.

Having a database which offers both sample precision annotations, various noise levels and two different ECG morphologies allows us then to present our new benchmarking score (JF) that equally accounts for all error types, including temporal inaccuracies, and is application-independent to truly assess different detector algorithms.

The combination of a precise ECG database with various noise levels and our new performance measure offers a tool to robustly benchmark new QRS complex detectors.

## Methods

### Glasgow University Database

The Glasgow University Database (GUDB) consists of two-minute two lead ECG recordings from 25 subjects each performing five different tasks, for a total of 125 records. The tasks were chosen to be repeatable and representative of common, realistic scenarios. The tasks were as follows:

- sitting

- using a tablet to perform a maths test

- walking on a treadmill

- using a hand-bike

- jogging

Ethical approval was given by the ethics committee at the Institute of Neuroscience and Psychology, School of Psychology at the University of Glasgow, with reference 300180026. The recruitment period was from October 15th, 2018 to December 21st, 2018.

Prior to the experiment, participants were given an information sheet and were asked to give signed consent by signing two consent forms, one for the researchers and another for

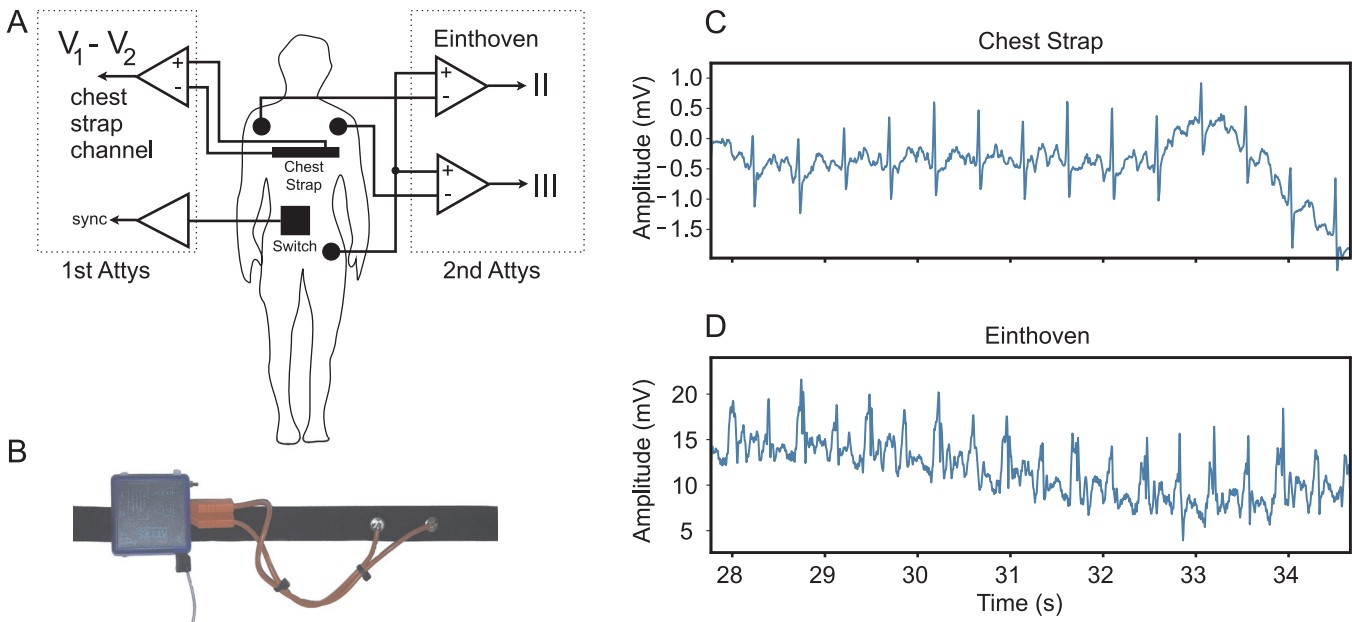

**Fig 3. Experimental setup and example ECG traces.** A: Wiring of the two wireless biosignal amplifiers (Attys). The first amplifier records $V_1 - V_2$ with a chest strap and the second amplifier Einthoven II & III with standard Ag/AgCl electrodes. B: Photo of the chest strap and the biosignal amplifier mounted on it. Signal comparison between C: chest strap and D: Einthoven II recording while the subject was jogging.

them to keep. Where the participant consented (24 out of the 25), a video synchronised to the data was recorded for each task. This video allows database users to see exactly how the movement was performed for each ECG recording and for any artefacts in the data to be identified. In addition, the acceleration of the torso was recorded. The participants were all over the age of 18 and had no known cardiovascular conditions. The database is available through the University of Glasgow's open access research data repository [17].

The ECG signals were recorded using an Attys Bluetooth data acquisition board (Glasgow Neuro LTD, Glasgow). This board has a sampling rate of 250 samples/sec and a resolution of 24 bits over a range of ±0.4 V. As this device is wireless, it increases electrical isolation and allows a moving subject to be recorded easily without the need of a cumbersome tether. The Attys features two differential analogue recording channels. Two separate Attys were used at the same time to record both Einthoven II & III and the approximate difference between the chest leads $V_1 - V_2$. These two configurations represent a best and worst-case recording setup. This allows the impact of recording setup on signal noise to be investigated.

1. The best-case setup uses the first Attys mounted on an elastic electrode chest strap (Amazon, UK), connected with short cables zip tied together (Fig 3A and 3B). This configuration minimises the effect of cable movement artefacts as much as possible and is worn tightly on the subject to prevent the electrodes from moving. As the chest strap is worn high around the chest, the electrodes are approximately in the same location as $V_1$ and $V_2$ in the standard six electrode chest configuration [18]. The left electrode (Pulse Medical, UK) on the chest strap is connected to the positive terminal of the differential amplifier and the right electrode is connected to the negative terminal. The second channel of the Attys records the switch signal used to synchronise the data with the video. The switch (Fig 3A) is worn on a belt around the waist. When switched, it produces an audible click and shorts channel two to ground. The circuit diagram configuration can be seen in Fig 3A.

2. The worse-case configuration uses the second Attys connected to standard ECG electrodes (Pulse Medical, UK) with loose cables. The positive terminal of the first differential amplifier is connected to the left hip, and the negative terminal to the right shoulder. The Attys is put into Einthoven ECG mode where the positive terminal of the CH1 differential amplifier is connected internally to the positive terminal of CH2 amplifier input. The negative terminal of the CH2 differential amplifier is connected to the left shoulder. This configuration allows two ECG signals to be recorded using only three cables. CH1 records Einthoven II between the left hip and right shoulder and CH2 records Einthoven III between the left hip and the left shoulder. The circuit diagram for this configuration can be seen in Fig 3A.

Having introduced the best and worse-case measurement situations, we show the corresponding raw signals in Fig 3C and 3D when the subject is jogging. The chest strap recording remains largely noise free while the Einthoven signal has significant noise contamination.

Once the subject had been successfully connected to the data acquisition equipment, the actual experiment followed the following protocol:

1. 120 second ECG recording, sitting down

2. 120 second ECG recording, timed maths questions on a tablet

3. 120 second break

4. 120 second ECG recording, walking on a treadmill at 2 kph

5. 120 second break

6. 120 second ECG recording, using a hand bike

7. 120 second break

8. 120 second ECG recording, jogging on a treadmill at 7 kph

9. Electrodes and chest strap removed from participant

The ECG data was stored without any signal processing in our open access ECG database [17]. When accessing the database for benchmarking 50 Hz and DC drift was removed on demand with a 4th order Butterworth notch filter and a 4th order 0.1 Hz highpass filter.

**Annotation procedure.** To annotate the data with heartbeat locations, a Python script was created which uses a Matplotlib [19] interactive plot. An ECG data file was loaded into the plot and ran through a heartbeat detection algorithm (EngZee segmenter from the BioSPPy library [20]) to provide an initial estimation of QRS locations. This estimation was then manually inspected by one of the authors (BP) to remove any false positives and add any missing QRS complexes. BP is a non-medical professional with training in signal processing and data analysis. Where there was too much noise to reliably annotate the entire recording, no annotation file was made. Of the 125 recordings, 2 chest strap and 19 loose cable recordings were unable to be annotated. This mostly occurred in the jogging scenario. The annotation sample locations are saved to a .tsv file when the plot is closed. This is performed for both the chest strap ECG and the Einthoven II loose cable recording.

## Detector software implementation

The eight algorithms chosen represent popular and well regarded QRS detectors, as well as a range of different detection techniques. The eight detector algorithms were: Pan & Tompkins [21], Elgendi et al [22], Kalidas & Tamil [8], Christov [23], Hamilton [24], the matched filter

detector, the EngZee Mod detector [25] and the WQRS detector [26]. The main criterion for selection was that the algorithm could be implemented in a real time system. All of the algorithms were implemented in Python, and the code can be found at Porr et al 2024 [27] and is continuously maintained.

## Evaluating detectors

**Traditional benchmarking method: Sensitivity.** All of the GUDB recordings that had corresponding annotations were evaluated: 123 for the chest strap and 106 Einthoven II setups. The widely used detection tolerance of $w = 10/f_s$ seconds was used [13] (see Fig 2). Note that all detectors will delay the signals to some extent, so the detected time stamps will always appear to be later. Thus, in order to allow sample precision evaluation, the median delay was subtracted from the true QRS detection time stamp and calculated separately for every subject, activity, and measurement protocol, as it was different each time.

The performance of the detector algorithms was compared using the current most commonly recommended performance parameter: sensitivity (Eq 4). The sensitivities for sitting & jogging, Einthoven & chest strap and the different detectors were then tested against a threshold of 90% with the help of the one sample t-test. Valid $p$ values need be based on at least 20 different sensitivity values. Alpha level was set to 0.05.

While the above approach simply assesses the sensitivity performance when QRS complexes occur either within a temporal window or not, they ignore the temporal jitter between the annotations and the QRS detections. In the next section, we present our new performance measure which also takes into account the jitter.

**New benchmarking measure: JF.** A new benchmarking approach JF is presented which takes into account the temporal jitter (J) of the detection point and combines it with all available statistical data for QRS detection which are extra beats (FP, false positives), missed beats (FN, false negatives) and true detections (TP, true positives) represented by the $F_1$-score (F) which is a performance measure of *the accuracy of the detection* [28]. Also, in contrast to the traditional approach, *no* fixed temporal window is used, but rather the $F_1$-score is penalised more and more with increasing jitter. Our new measure JF is open source (code: [29]).

Fig 4 demonstrates the working principle of the JF algorithm which is now explained step by step. Its input is the sample points $d_k$ of a QRS detector which are compared against the QRS detection annotations $a_l$. As a first step, we need to compensate for constant detection delays by subtracting the median delay between the detections $d_k$ and annotations $a_l$ from the individual detection timestamps:

$$\tilde{d}_k = d_k - \text{median}\left(\forall l, \min_k(|a_l - d_k|)\right) \tag{8}$$

where $\forall$ means "for all" with $l = 0 \ldots \text{card}(a_l) - 1$.

As a next step, we now need to find the mapping between the annotations and the detections to be able to identify true detections, missed QRS complexes and spuriously detected QRS complexes. In order to achieve this, we iterate through all annotation indices $l = 0 \ldots \text{card}(a_l) - 1$ and find, for each annotation timestamp $l$, the index $k$ within the detections $\tilde{d}_k$ which has the smallest temporal difference and store it in the set $p_l$:

$$p_l = \text{argmin}_{k=0\ldots\text{card}(\tilde{d}_k)-1}|a_l - \tilde{d}_k| \tag{9}$$

where argmin outputs the index $k$ where the minimum between $|a_l - \tilde{d}_k|$ occurs. Without any missing detections or spurious detections, the index of $p_l = l$ will simply be a one-to-one

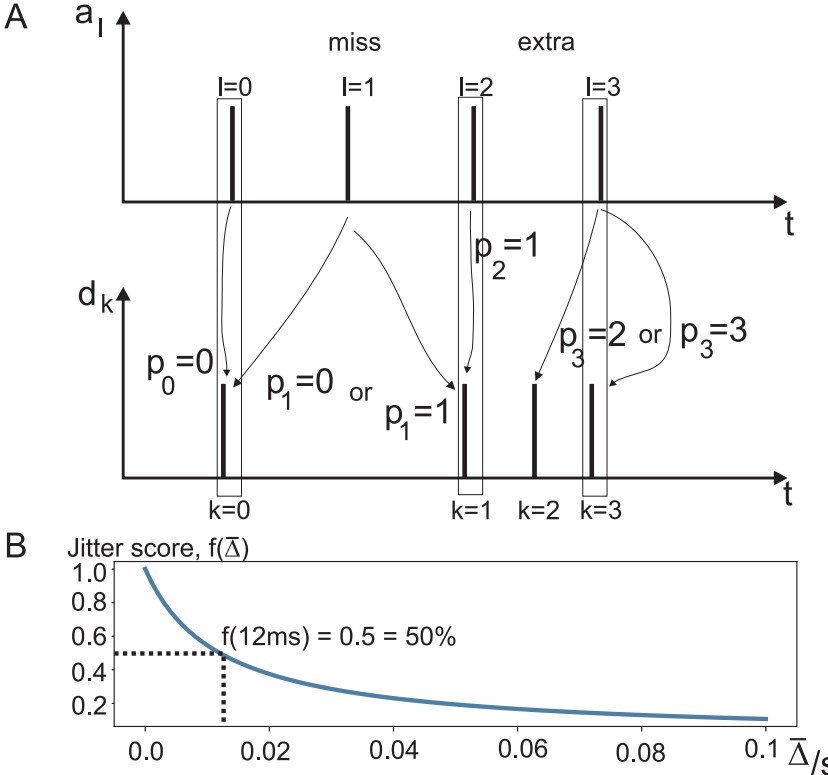

**Fig 4. Graphical illustration of the JF algorithm.** A: Time stamps $a_l$ of the QRS annotations versus the time stamps of the actual QRS detections $d_k$. $p_l$ links the index numbers of the annotations $l = 0, 1, 2, 3$ to the index numbers $k$ of the detections $d_k$. B: Mapping function Eq 11 which maps the average jitter $\bar{\Delta}$ to a score between zero and one which reaches 0.5 at a jitter of 12 ms.

mapping between annotations and detections. However, as illustrated for the annotation $l = 1$ in Fig 4, there is no matching detection, and thus it is a missing beat. Depending if the annotation is closer to $d_0$ or $d_1$, the annotation will be falsely associated with one of them. This can be rectified by switching the perspective and then iterating through the $p_l$ indices to check if there are duplicates. For example, here $p_0 = p_1 = 0$ both point to the same detection. However, clearly $p_0$ is closer to $d_0$ so that $p_1 = 0$ should be discarded. Similarly, if the annotation at $l = 1$ were closer to $k = 1$, then $p_1 = p_2 = 1$ is the duplicate position. Thus, formally we minimise the temporal difference between the detection indices while keeping the correspondence between QRS annotation and detection unique:

$$\Delta_m = \left\{ \frac{\min_k |a_l - d_{p_k}|}{fs} \text{ with unique } p_k \right\} \tag{10}$$

where $\Delta_m$ are now the jitter values of true detections, and the number of elements card($\Delta_m$) are the number of the true positive detections.

As a next step, the jitter $\Delta_m$ needs to be evaluated and turned into a performance measure between 0 and 100%. We define the following mapping function as the jitter score:

$$f(\bar{\Delta}) = \frac{1}{1 + \frac{\bar{\Delta}}{12 \text{ ms}}} \tag{11}$$

where $\bar{\Delta}$ is the average over all individual jitter values $\Delta_m$. This means that an average jitter of $\bar{\Delta} = 12$ ms results in a performance of 0.5 or 50%.

Having calculated the average temporal jitter, we now need to calculate the $F_1$-score. Here, the number of the individual jitter readings card($\Delta_m$), being the number of the true positive (*TP*) detections, can be used to calculate the number of missed beats (false negatives = *FN*) and extra beats (false positives = *FP*):

$$TP = \text{card}(\Delta_m) \tag{12}$$

$$FP = \text{card}(d_k) - \text{card}(\Delta_m) \tag{13}$$

$$FN = \text{card}(a_l) - \text{card}(\Delta_m) \tag{14}$$

Having obtained the statistical parameters, we can now calculate the $F_1$-score:

$$F_1 = \frac{2TP}{2TP + FP + FN} \tag{15}$$

The $F_1$-score is a normalised measure between zero and one. As a final step, we can now combine the jitter score $f$ (Eq 11) and the $F_1$-score (Eq 15):

$$\text{JF} = F_1 \cdot f(\bar{\Delta}) \cdot 100\% \tag{16}$$

JF reaches 100% if the average jitter is zero ($\bar{\Delta} = 0$) and there are no false negative or false positive detections. Both increase of jitter and increase of false negative or false positive detections will decrease the value of JF.

The JF performance values (Eq 16) for sitting and jogging in combination with Einthoven, chest strap and the different detectors are then tested against a threshold of 90% with the help of the one sample t-test. Valid *p*-values need be based on at least 20 different sensitivity values. Alpha level is set to 0.05.

## Results

### Traditional method: Sensitivity

Fig 5 shows sensitivity values for the different detectors as calculated with Eq 4. The asterisk indicates if the sensitivity of a detector is significantly greater than 90%. It can be seen that 22 of the 32 sensitivity values are above 90%. Panel A shows the results for the standard Einthoven II leads for sitting and jogging. Let us first focus on sitting with the Einthoven leads. This is a standard condition comparable to other noise free databases such as the MITDB with virtually no artefacts. Except for the Hamilton detector, all other detectors reach significantly sensitivity values above 90%. Jogging creates movement artefacts through cable movements and muscle noise (see Fig 3) which then cause sensitivity values for more detectors to significantly drop below 90% such as Elgendi, matched filter, EngZee, Hamilton and WQRS. Fig 5B shows the same detectors for sitting and jogging but with a chest strap. This reduces the movement artefacts and muscle noise but at the same time, the ECG measured at the chest electrodes has a different shape compared to the standard Einthoven leads. Still, 12 sensitivity values are all significantly above the 90% threshold and it makes it hard to decide which detector to choose. Overall the large detection error margin between detector timestamp and ground truth results in very high sensitivity readings.

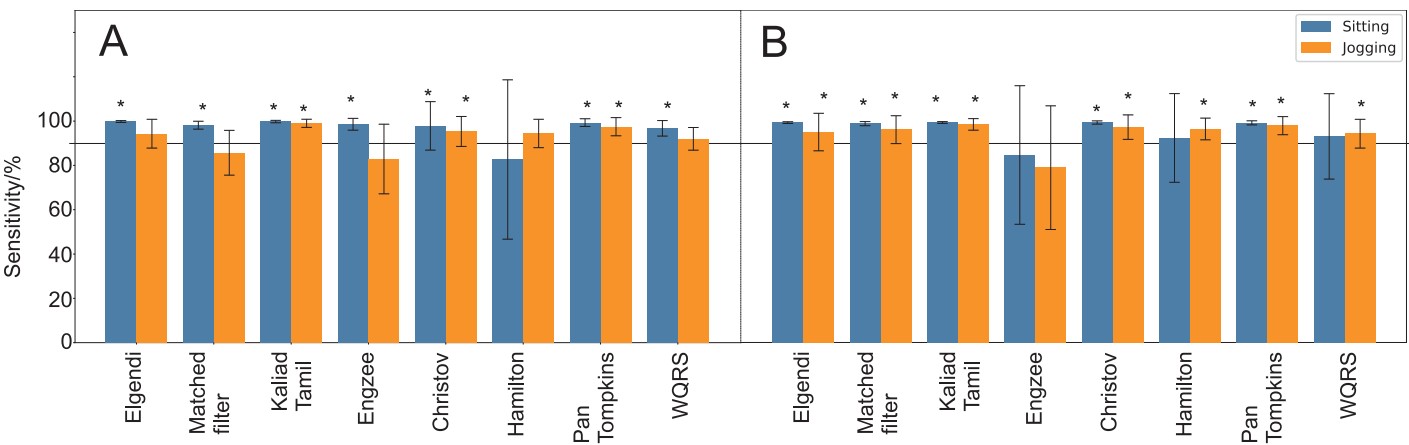

**Fig 5. Classical sensitivity values.** A: Einthoven, B: Chest strap. Significantly above 90% indicated with an asterisk.

## JF

Fig 6 shows the JF score for the different detectors as calculated with Eq 16. Overall it is qualitatively apparent that the introduction of jitter to the performance measure causes a much larger variation of the JF values than the sensitivity above (Fig 5). Let us start again with the Einthoven leads while sitting which has the best signal to noise ratio. Now, only the EngZee, the Elgendi and the WQRS detector reach JF values significantly above 90%. When using a chest strap instead, the Elgendi detector and the matched filter reach significantly JF values above 90%. This shows that the Elgendi detector is robust against different ECG morphologies and keeps its excellent performance irrespective of lead configurations.

When jogging, the signal to noise ratio of the ECG signal drops because of Electromyogram (EMG) and movement artefacts. Here, clearly the JF score drops which pushes all detectors significantly below the 90% threshold. Note that jogging is the worst case scenario and that the other experiments introduce gradually more and more noise.

Fig 7 benchmarks the detectors against increasing noise levels imposed on the ECG. We have taken the 3 detectors EngZee, Elgendi and the WQRS which had excellent performance in the noise free condition (Fig 6) and subjected them to noise. In addition, we have taken the

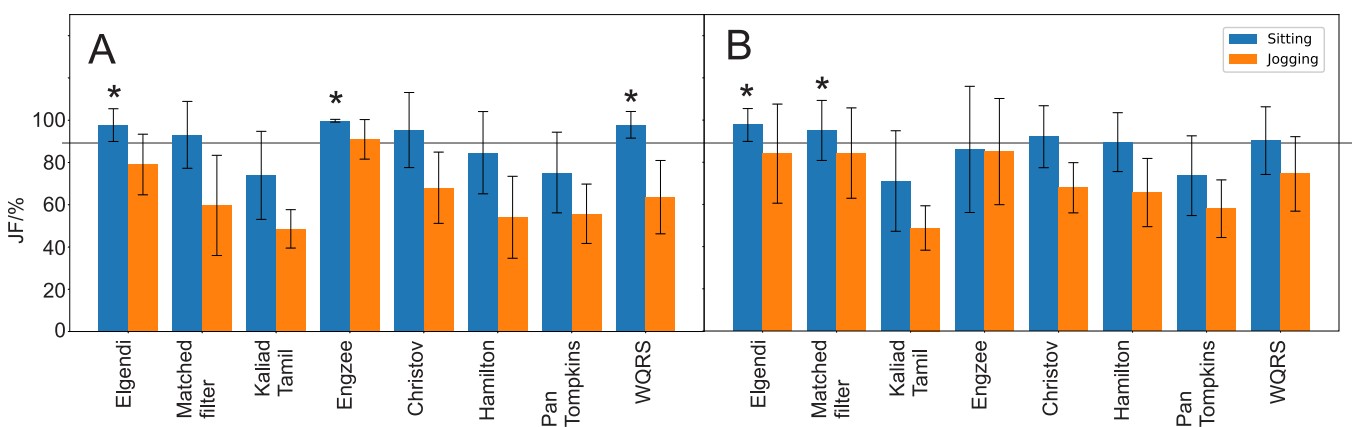

**Fig 6. JF values.** A: Einthoven, B: Chest strap. Significantly above 90% indicated with a "*".

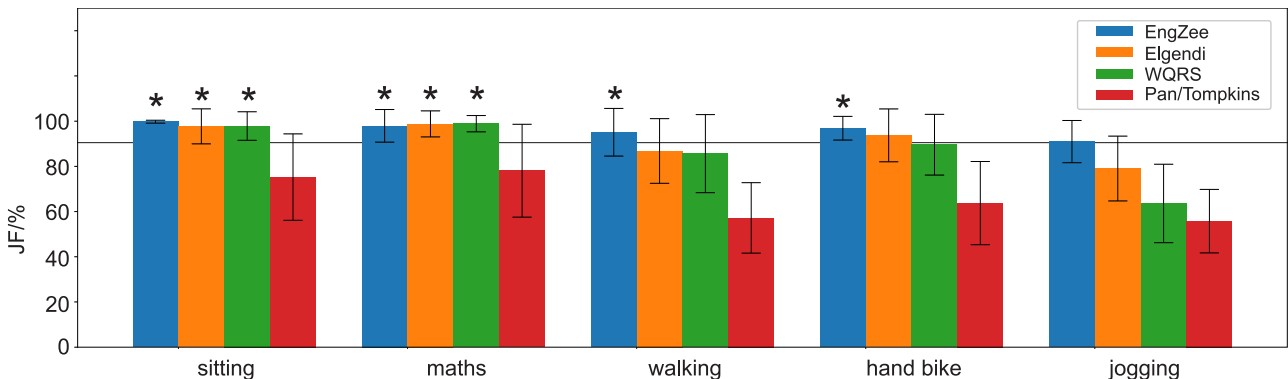

**Fig 7. Comparing detector performance against increasing signal to noise ratios.** The JF benchmark value was calculated for the different activities of sitting, doing a maths test, walking, operating a hand-bike and jogging. The 4 detectors EngZee, Elgendi, WQRS and Pan& Tompkins have been benchmarked. If a JF value is significantly above 90% it is indicated with an asterisk.

worst performer, namely the the Pan & Tompkins detector. The maths test introduces small and irregular noise through movement artefacts and muscle noise. We note that all three contemporary detectors EngZee, Elgendi and WQRS perform identically here. As mentioned before, the EngZee detector undertakes a final maximum detection of the QRS-complex which is no longer advantageous in the presence of noise, possibly because of multiple peaks introduced by high frequency EMG noise. Nevertheless, all three detectors (excluding Pan & Tompkins) are significantly above 90%. During walking and while operating a handbike, the performance of the Elgendi and WQRS detectors drops significantly below 90%. Both activities generate stronger artefacts. The handbike generates more upper body artefacts while walking generates artefacts from muscles in the whole body. Jogging is the worst case scenario where sometimes, even through visual inspection, QRS complexes simply cannot be identified. Here, all detectors are significantly below the 90% mark and thus not reliable enough, certainly for high precision tasks such as heartrate variability. The EngZee detector is the clear winner for noisy ECGs while for low-noise tasks such as sitting in a chair, EngZee, Elgendi and the WQRS perform equally well, producing significantly JF values above 90%.

## Discussion

In this paper, we have presented a new approach to benchmarking QRS detectors. In order to achieve this we created first a new ECG database [17] with real world noise, artefacts and two popular ECG recording approaches reflecting both standard clinical practice and lifestyle applications. Secondly, we developed a new benchmarking score for QRS detectors (JF) which combines the $F_1$- score with the temporal Jitter of the detection. To demonstrate how our new score can be used to benchmark QRS detectors, we implemented eight popular QRS detection algorithms in Python [27].

We will now discuss other benchmark measures and also which ECG databases they used for benchmarking. Friesen et al [31] used as a benchmark measure the percentage of QRS complexes correctly detected (TP) and the number of false positives (FP). However, they did not take into account the number of missed beats (FN). While this would allow the calculation of the positive predictability they simply listed the TP and FN values in tables. Temporal jitter was not taken into account. However, they noted the lack of noisy ECGs in the MIT-BIH database making it not directly suitable for benchmarking. They solved this problem by artificially adding EMG-noise, powerline interference, baseline shifts and a combination of them.

However, adding artificially created noise causes a rather uniform noise distribution over all subjects, while in our case the noise is individual and varies between subjects. In addition, our noise originates from real experiments while artificially generated noise relies on a good noise model but this is hard to achieve, in particular for EMG and movement artefacts. In fact, movement artefacts are modelled in this paper as a simple sudden baseline shift.

The work by Rodrigues et al [32] uses as a benchmark measure the percentage of missed detections (FN) and false alarms (FP). As with the work discussed above [31], these statistical measures are not combined but shown separately. Instead of using the MIT-BIH database, a special ECG database was created from patients fitted with a pacemaker. Again, it was noted that the ECG recordings did not contain any noise but instead of adding noise artificially, real noise was recorded and then added to the noise-free ECGs. This work acknowledges that noise cannot be created easily artificially but needs to be recorded to be realistic, which is in the same spirit as our work. However, the noise was nevertheless then added to the recordings artificially, and did not emerge from a recording made during different tasks. Adding noise artificially allows a gradual reduction of the signal to noise ratio but it assumes that the nature of the noise stays the same. Instead, we decreased the signal to noise ratio successively by having different tasks with increasing noise levels.

Sensitivity, for example used by Merah et al [35], is probably the most popular performance measure and is referenced in most papers. The problem with sensitivity is that it only takes into account the true positives (TP) and false negatives (FN) but not the false positives (FP). For that reason, this paper also calculates the positive predictive value (PPV), which is then presented side by side with the sensitivity.

Merino et al [36] benchmarked the QRS detection through sensitivity, positive predict value and accuracy. The latter was defined as $A = TP/(TP + FP + FN)$ (s.i.c) in contrast to the textbook definition of accuracy (Eq 6)–roughly reminiscent of the $F_1$ score. Instead of using just the MIT-BIH database, the authors randomly selected ECGs from five Physionet databases to gain a higher variability of the data in terms of ECG morphologies and recording conditions but did not add any noise to the recordings. Consequently, all results are above 99% with small variations effectively being meaningless which ultimately has motivated us to create our noisy ECG database so that sensitivity values are not all between 99% and 100%. For other recent QRS detection algorithms and their resulting sensitivities, see for example Kumar et al [37] where all tabulated sensitivities range from 99.31%. . .99.98%, again making it impossible to perform any meaningful statistical comparison.

Given the centre stage of sensitivity, we now compare sensitivity values obtained from our dataset with those from the original literature. The QRS detectors used here are those which are most popular and are actively maintained on github as an open source repository. Apart from being popular, they also provide the different basic concepts for real time ECG detection, upon which newer detectors build.

Table 1 shows the sensitivity results for the eight QRS detectors. The 2nd column states the originally cited sensitivities from the literature. Note that none of the eight papers specify the width of the temporal window $w$ (Eq 7) and thus the temporal detection tolerance is declared as "unspecified". As with the studies mentioned above, not all used the MIT/BIH database so that in the detailed discussion below, we will state the dataset used for every detector—if known. The 3rd column shows our sensitivity results using the standard temporal window of $w = 10/f_s$ (Eq 7) using our new GUDB database from the Einthoven leads while sitting. The 4th column shows the results of our new JF performance measure. In total, 3911 QRS complexes were used to calculate the average and standard deviations of sensitivities and JF scores for all detectors.

**Table 1. Original cited sensitivities from the literature, in comparison to the new overall JF Benchmark scores for each detector using GUDB.**

| | Original cited sensitivity (%) (* based on RR intervals: see main text) | Sensitivity and standard deviation of Einthoven II, sitting, based on GUDB and a tolerance of $w = 10/f_s$ | JF score and standard deviation of Einthoven II, sitting, based on GUDB |
|---|---|---|---|
| Elgendi et al. [30] | 92.66...98.31 | 99.4 ± 0.4 | 97.7 ± 7.7 |
| Kalidas & Tamil [8] | 99.88 | 99.4 ± 0.6 | 73.9 ± 20.8 |
| Engzee [25] | 96.50* | 98.2 ± 2.7 | 99.8 ± 0.6 |
| Christov [23] | 84.5...96.5 | 97.4 ± 10.9 | 95.4 ± 17.6 |
| Hamilton [23] | 99.74...99.80 | 82.3 ± 35.8 | 84.6 ± 19.5 |
| Pan & Tompkins [21] | 99.30 | 98.9 ± 1.7 | 75.3 ± 19.1 |
| Matched filter | N/A | 97.7 ± 1.8 | 93.3 ± 15.9 |
| WQRS [26] | 99.65 | 96.3 ± 3.5 | 97.8 ± 6.3 |

Pan & Tompkins [21] were one of the first to develop a real-time QRS detection algorithm. The performance was analysed by playing tapes of ECG recordings from the MIT/BIH database which were turned into digital signals, processed in a digital processor and then turned back into analogue signals for comparison with the annotations. Coincidence between the detected QRS complexes and the annotations was assessed by visual inspection so that a precise jitter tolerance could not be given. Overall, Pan & Tompkins [21] reported a sensitivity of 99.3% (Table 1) which we have confirmed closely. However, our JF benchmark produces a value of 75.3±19.1 which is not only significantly below the 90% margin but also has a huge standard deviation (see also Fig 7). The Pan & Tompkins algorithm is the oldest detector benchmarked here but its adaptive thresholding is still used in newer algorithms as outlined below.

Instead of bandpass filtering the ECG, Kalidas & Tamil [8] employed the stationary wavelet transform to filter the ECG which resulted in a high sensitivity in their paper of 99.88% (Table 1) which is reproduced by our work here. However, neither a statistical significance test nor the detection tolerance (see Eq 7) is published. Moreover, given their 99.88% sensitivity, it suggests that again a large temporal jitter $w$ was permitted. However, the exact margin is not mentioned in the text. While the sensitivity does not take jitter into account, our JF benchmark does: indeed, it is one of the lowest (Fig 6) for both Einthoven and chest strap and is comparable to the original Pan & Tompkins detector. This comes as no surprise because Kalidas & Tamil also used the original Pan & Tompkins thresholding but preceded it by a wavelet filter instead of a bandpass filter. A wavelet filter is essentially a bandpass filter so that the overall performance is very similar to the Pan & Tompkins detector. However, while Kalidas & Tamil pioneered ECG detection with wavelet filters, this field has moved on. Wavelets with their matching scaling function allow the creation of filter cascades and then the recombination of the resulting signals for thresholding [34, 38]. We acknowledge the development of this field over the years, but we have kept to the original work of Kalidas and Tamil as newer wavelet analyses are beyond the scope of this paper.

Hamilton [24] had in mind an open source QRS detector with a high sensitivity which is a further development of the detector proposed by Pan & Tompkins [21]. In order to be able to run it on a microcontroller, Hamilton [24] presented a stripped down version of their full detector implementation, omitting certain QRS detection rules. For Hamilton's original detector, the author reported a sensitivity of 99.74% and for the microcontroller version 99.80% (Table 1). The sensitivities were not compared to other detectors nor has a statistical analysis

been performed. The author stated that the detectors perform comparably. The jitter tolerance is not mentioned in the paper. However, we report a sensitivity of just 82.3 ± 35.8% which falls substantially short of the claimed 99.74%. . .99.80%. This again shows that the Pan & Tompkins detection algorithm cannot be much improved beyond its original performance.

Instead of improving the detection algorithm, the work by Elgendi et al [22] aimed to optimise the bandpass filter frequencies used for the QRS detection. Different frequency bands were benchmarked and the 8 − 20 Hz band was chosen to be optimal, having the highest sensitivity. However, all originally cited sensitivities, even the sub-optimal ones, vary only between 92.66% and 98.31% (Table 1) where some deviate less than one percent. There is no statistical analysis in the paper stating which band is significantly better than another. A temporal tolerance is not given. The lack of statistical tests in the original paper and the omission of the tolerance render the recommendations towards cut-off frequencies questionable. While their own analysis is inconclusive, our own JF benchmark (Table 1) places them into the category of the best performers and is certainly an algorithm which could be used in production.

Christov [23] starts off not just with Einthoven I but creates a complex lead combining the derivatives from all standard 12 leads. This is then sent through various thresholds to detect the QRS complexes. The author achieved a sensitivity of 84.5%. . .96.5% at a tolerance of $w$ = 60 ms (Table 1). The article has no discussion section and thus does not compare its results to other approaches and has no statistical evaluations. In the conclusion section, the author claims that: "The statistical indices are higher than, or comparable to, those cited in the scientific literature." [23]. This is not confirmed for our JF measure where, even when sitting, its value is significantly below 90% with also a very high standard deviation.

Similar to Christov [23], taking the derivative of the ECG signal is the central idea of Englese et al [37] which is then turned into a real-time version by Lourenco et al [25] using an adaptive threshold. The paper then compares the real-time version with the previous off-line version and the detector by Christov [23] which requires a fixed threshold. This paper does not use the MIT-DB database but used locally recorded ECGs. The author also devised a different performance measure: the deviation from the mean RR interval needs to be less than 10% for individual RR pairs, or in other words, the RR interval variation needs to be within normal limits of the resting heart rate variability [38]. The author's online algorithms leads to average valid RR intervals of 96.50% (Table 1) depending upon on- or offline algorithms, use of electrodes, the algorithm itself and filtering. These high readings are expected because the performance measure will most likely detect only crude deviations from the mean RR interval, for example a missed beat, which then results in twice the RR interval being measured, or an additional spurious detection, which results in a very short RR interval. Even more important is that this measure does not compare against the ground truth of RR intervals and only looks at the self consistency of the RR intervals, which might have been wrong in relation to the annotations in the first place. There has been no statistical test to determine which of these results differ significantly, but one could easily have been performed with the reasonably large number of subjects available ($N$ = 62). Our JF benchmark produces a high JF value but with a low standard deviation.

Overall, virtually every paper reporting very high sensitivities of 98% or more is not at all helpful, and has only been possible because of high temporal tolerances of probably 100 ms or even more. Our new JF benchmark, taking into account both the all-or-nothing errors such as missed beats and the continuous jitter, produces values which allow meaningful comparisons between detectors. We conclude that of the ones benchmarked EngZee, Elgendi and the WQRS are the best detectors available. However, our JF benchmark can be applied to any new detector developed and will inform the development of new QRS detectors which result in ultimately more reliable clinical equipment. We have also shown an approach how to create an

ECG database with increasing levels of noise. So far our database is only from healthy subjects but calls for a similar approach for patients with a pace maker or pathological heart conditions.

## Author Contributions

**Conceptualization:** Bernd Porr.

**Data curation:** Bernd Porr.

**Formal analysis:** Bernd Porr.

**Investigation:** Bernd Porr.

**Methodology:** Bernd Porr.

**Project administration:** Bernd Porr.

**Resources:** Bernd Porr.

**Software:** Bernd Porr.

**Supervision:** Bernd Porr.

**Validation:** Bernd Porr.

**Visualization:** Bernd Porr.

**Writing – original draft:** Bernd Porr.

**Writing – review & editing:** Bernd Porr, Peter W. Macfarlane.

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
