## [Decision Letter · Decision Letter 0]

2 Jan 2024

PONE-D-23-23819A new QRS detector stress test combining temporal jitter and accuracy (JA) reveals significant performance differences amongst popular detectorsPLOS ONE

Dear Dr. Porr,

Thank you for submitting your manuscript to PLOS ONE. After careful consideration, we feel that it has merit but does not fully meet PLOS ONE’s publication criteria as it currently stands. Therefore, we invite you to submit a revised version of the manuscript that addresses the points raised during the review process.

Reviewers have suggested several revisons. Authors are enocuraged to revise upon the comments and submit again.

We look forward to receiving your revised manuscript.

Kind regards,

Noman Naseer, PhD

Academic Editor

PLOS ONE

Journal Requirements:

“B.P. is CEO of Glasgow Neuro LTD which manufactures the Attys DAQ board.”

We note that one or more of the authors are employed by a commercial company: Glasgow Neuro LTD

Additional Editor Comments (if provided):

Reviewer's have suggested several revisons. Authors are enocuraged to revise upon the comments and submit again.

Reviewers' comments:

Reviewer's Responses to Questions

**Comments to the Author**

1. Is the manuscript technically sound, and do the data support the conclusions?

Reviewer #1: Yes

Reviewer #2: Partly

Reviewer #3: Yes

2. Has the statistical analysis been performed appropriately and rigorously? 

Reviewer #1: Yes

Reviewer #2: No

Reviewer #3: No

3. Have the authors made all data underlying the findings in their manuscript fully available?

Reviewer #1: Yes

Reviewer #2: Yes

Reviewer #3: No

4. Is the manuscript presented in an intelligible fashion and written in standard English?

Reviewer #1: Yes

Reviewer #2: No

Reviewer #3: Yes

5. Review Comments to the Author

Reviewer #1: However, some problems still exist, and the manuscript need to be improved by considering the following comments:

1. The Abstract is clear and comprehensive.

2. The figures are clear.

3. Key terms of equations must be defined.

4. Give more detaile aboute the Dataset. Also, the authors have specified the reasons for selecting the dataset.

5.The results section is clear.

6. The authors should mention to "where the proposed system is applicable".

7.The references should be the last five years.

Reviewer #2: The aim of this paper is to demonstrate a new quality measure for the evaluation of QRS detectors. The proposed measure combines temporal jitter with accuracy, which makes it possible to assess the impact of all errors that arise during the detection of QRS complexes, especially in the presence of noise caused by patient movement. In this study, six popular QRS detectors were compared taking into account the proposed IAM quality index. The Glasgow University Database (GUDB) which consists of two-minute two lead ECG 100 recordings from 25 subjects each performing five different tasks, for a total of 125 records, was used for comparison.

The paper is interesting and addresses a problem with clinical applicability. However, I recommend rearranging the manuscript in relation to the research steps, for a better fluency of the information presented.

Thus, overall, the structure of the paper is difficult to follow in relation to the dedicated sections.

The proposed methodology design should be more clarified.

1. The presentation of the database and of the performance measures for detection efficacy should be presented in experimental setup.

2. Computation of accuracy depends on the knowledge of the true negative (TN), namely (equation 15):

Accuracy = (TP + TN) / (TP + TN + FP + FN)

However, in the context of QRS detection, the concept of true negatives is not applicable. The authors currently use an alternative definition of the accuracy which does not include the true negative. However, these alternative definitions do not conform to the standard definitions provided in the literature, and their interpretation is unclear. Therefore, this indicator should be removed from the manuscript or rearrenged.

3. Table 1. The description of the table suggests that all results presented are from the GUDB database, whereas the first column contains results obtained from the MIT-BIH AR database, which is not mentioned in any way and is misleading. Furthermore, what is meant by the phrase unspecified tolerance (%). In the following columns, it is specified that these are results obtained (I think) from the work on the basis of signals from GUDB records. However, combining the results in this way may mislead readers. Another inaccuracy is the value given after the plus/minus sign. What is this value? How was it determined? In addition, Table 1 of Hamilton's paper is assigned the wrong bibliographic number, 31 instead of 24.

4. The Discussions should be oriented towards summarizing the results and comparing them with those of similar studies and presenting the advantages and novelty of the proposed algorithm. In addition, the limits of the research carried out should be identified and briefly presented. Nevertheless, are important to mention the perspectives for the application in clinical practice of the new algorithm, and new research directions derived from the research carried out.

5. There is no conclusion section.

6. No translation of abbreviations, e.g. EMG.

7. What is the relevance to the manuscript of the information that the Pan&Tompkins algorithm was originally written for the Z80 microprocessor?

8. There is a lot of formulations in the paper that are not supported by the results (e.g. line 353-354).

9. No description of the numerical experiment in which a standard deviation value of 35.8% is obtained (line 367). How many times were the calculations performed?

10. It seems to me that for the quality of the work as a whole, it would be advisable to use the MIT-BIH AR database and then compare the results. By which the work would be undoubtedly better appreciated.

11. Line 172: The paper says that the annotation of heartbeats in ECG signals was performed with a Python script using an interactive plot. Were the determined QRS positions verified by an expert cardiologist? From the information available, it appears that the development of this software ended in 2018/2019 and furthermore, no information is provided about the signal processing used.

Reviewer #3: This paper presents an ECG testing database with precise annotations under realistic noise levels and a new benchmarking score (JA) that equally accounts for all error types, including temporal inaccuracies, and is application-independent to truly assess different detector algorithms. The paper idea is interesting. But there are many issues in the paper that need further clarification:

1. The contribution of the paper needs to be properly specified in the abstract.

2. The figures should be place with in the text for better readability of the article during review process.

3. What are the key technical problems that this paper tries to solve and what are the drawbacks of the existing QRS detectors?

4. In the introduction, I suggest that the author rearrange the introduction section.

5. The paper needs to discuss recent developed QRS detection methods and include in the comparison. Some of the recent methods are in the following works:

Design of A Biorthogonal Wavelet Transform Based R-Peak Detection and Data Compression Scheme for Implantable Cardiac Pacemaker Systems

Efficient QRS complex detection algorithm based on Fast Fourier Transform

Design of Efficient Fractional Operator for ECG Signal Detection in Implantable Cardiac Pacemaker Systems

Hardware Emulation of a Biorthogonal Wavelet Transform Based Heart Rate Monitoring system

6. The availability of dataset should be provided in the paper before reference section via link.

7. Why post processing is required.

8. Is the annotation was done through software or by medical practitioners.

9. Discuss the computational complexity of proposed model and compare with the existing methods.

10. There are lots of typo error and grammatical error. That should be corrected.

6. PLOS authors have the option to publish the peer review history of their article (what does this mean?). If published, this will include your full peer review and any attached files.

Reviewer #1: **Yes: **Ali Noori kareem

Reviewer #2: No

Reviewer #3: No

---

## [Decision Letter · Decision Letter 1]

19 Aug 2024

A new QRS detector stress test combining temporal jitter and F-score (JF) reveals significant performance differences amongst popular detectors

PONE-D-23-23819R1

Dear Dr. Porr,

We’re pleased to inform you that your manuscript has been judged scientifically suitable for publication and will be formally accepted for publication once it meets all outstanding technical requirements.

Kind regards,

Noman Naseer, PhD

Academic Editor

PLOS ONE

Additional Editor Comments (optional):

All concerns of reviewers have been addressed.

Reviewers' comments:

Reviewer's Responses to Questions

**Comments to the Author**

1. If the authors have adequately addressed your comments raised in a previous round of review and you feel that this manuscript is now acceptable for publication, you may indicate that here to bypass the “Comments to the Author” section, enter your conflict of interest statement in the “Confidential to Editor” section, and submit your "Accept" recommendation.

Reviewer #4: All comments have been addressed

2. Is the manuscript technically sound, and do the data support the conclusions?

Reviewer #4: Yes

3. Has the statistical analysis been performed appropriately and rigorously? 

Reviewer #4: Yes

4. Have the authors made all data underlying the findings in their manuscript fully available?

Reviewer #4: Yes

5. Is the manuscript presented in an intelligible fashion and written in standard English?

Reviewer #4: Yes

6. Review Comments to the Author

Reviewer #4: Overall, the manuscript does seem well-put together now, particularly after the revisions. They have established well-defined parameters to gauge JF in their draft. I am happy with the amendments they have made and would recommend it for publication now.

7. PLOS authors have the option to publish the peer review history of their article (what does this mean?). If published, this will include your full peer review and any attached files.

Reviewer #4: No

---

## [Editor Report · Acceptance letter]

23 Oct 2024

PONE-D-23-23819R1 

PLOS ONE

Dear Dr. Porr, 

I'm pleased to inform you that your manuscript has been deemed suitable for publication in PLOS ONE. Congratulations! Your manuscript is now being handed over to our production team.

Kind regards, 

on behalf of

Dr. Noman Naseer 

Academic Editor

PLOS ONE